



# Composition and oxidation state of sulfur in atmospheric particulate matter

Amelia F. Longo[1], David J. Vine[2], Laura E. King[1], Michelle Oakes[3], Rodney J. Weber[1], L. Gregory Huey[1], Armistead G. Russell[1], Ellery D. Ingall[1]

5   [1]School of Earth and Atmospheric Sciences, Georgia Institute of Technology, 311 Ferst Drive, Atlanta, GA 30332-0340, USA
[2]Advanced Photon Source, Argonne National Laboratory, 9700 S. Cass Avenue, Argonne, IL 60439, USA
[3]Tennessee Department of Environment and Conservation, Division of Air Pollution Control, Nashville, TN 37206, USA

10   *Correspondence to:* Ellery Ingall (ingall@eas.gatech.edu)





**Abstract.** The chemical and physical speciation of atmospheric sulfur was investigated in ambient aerosol samples using a combination of Sulfur Near-Edge X-ray Fluorescence Spectroscopy (S-NEXFS) and X-ray fluorescence (XRF) microscopy. These techniques were used to determine the composition and oxidation state of sulfur in common primary emission sources and ambient particulate matter collected from the greater Atlanta area. Ambient particulate

5    matter samples contained two oxidation states: $S^0$ and $S^{+VI}$. Individual particles (> 1 µm) contain $S^0$ in 95% of the individual aerosol particles analyzed. Linear combination fitting revealed that $S^{+VI}$ in ambient aerosol was dominated by ammonium sulfate as well as metal sulfates. The finding of metal sulfates provides further evidence for acidic reactions that solubilize metals such as iron during atmospheric transport. Emission sources including biomass burning, coal fly ash, gasoline, diesel, volcanic ash, and aerosolized Atlanta soil, and the commercially available

10    bacterium *Bacillus subtilis* contained only $S^{+VI}$. A commercially available *Azotobacter vinelandii* sample contained approximately equal proportions of $S^0$ and $S^{+VI}$. $S^0$ in individual aerosol particles most likely originates from primary emission sources such as aerosolized bacteria or incomplete combustion.





## 1 Introduction

Sulfur (S) in atmospheric aerosols has garnered significant interest because of its influence on environmental processes (Bufalini, 1971;Galloway, 1995;Van Grieken et al., 1998) and human health (Burnett et al., 1995). S-rich aerosols are highly hydroscopic and can serve as cloud condensation nuclei. On a global scale, cloud condensation

nuclei participate in cloud formation processes to change atmospheric albedo (Charlson et al., 1987;Quinn and Bates, 2011), which can ultimately influence global climate. Atmospheric sulfur is responsible for the production of sulfuric acid, which has been implicated in the solubilization of metals in aerosol (Wiederhold et al., 2006;Nenes et al., 2011;Sullivan et al., 2007;Oakes et al., 2012a). Solubilization of recalcitrant mineral phases during atmospheric transport can release nutrient elements, such as iron, which impacts biological productivity in many ocean regions

(Longo et al., 2014;Mahowald et al., 2005;Jickells et al., 2005). Additionally, soluble metals present in the urban environment may cause adverse respiratory events among affected populations (Fang et al., 2016;Pardo et al., 2016).

The range of sulfur oxidation states as well as organic and inorganic forms present in typical samples confound characterization of aerosol sulfur. Characterization approaches including ion chromatography, X-ray fluorescence, and inductively coupled plasma mass spectrometry have been used to quantify bulk elemental or bulk

concentrations, which can be used to infer the composition of sulfur in aerosols. However, these techniques cannot determine the oxidation state or directly identify the chemical form of aerosol sulfur. Several studies have demonstrated the effectiveness of synchrotron-based sulfur X-ray absorption near-edge structure (XANES) and the closely related sulfur near-edge X-ray fluorescence spectroscopy (NEXFS) to determine the oxidation state and chemical form of sulfur in environmental samples (Solomon et al., 2003;Prietzel et al., 2011;Morra et al., 1997;Farges

et al., 2009); however, the application of these techniques to ambient aerosol samples has largely been limited to studies characterizing a few ambient aerosol samples that aim to explore the feasibility and limits of applying XANES and NEXFS to aerosol sulfur studies (Takahashi et al., 2006;Pongpiachan et al., 2012a;Pongpiachan et al., 2012b;Higashi and Takahashi, 2009;Cozzi et al., 2009).

Consistent with traditional techniques, synchrotron-based XANES studies indicate that sulfur occurs

primarily in the $S^{+VI}$ oxidation state, largely, as inorganic sulfate compounds. Primary emission sources of $S^{+VI}$ include both anthropogenic and biogenic sources, including combustion of fossil fuels, biomass burning, volcanoes, and sea spray (Querol et al., 2000). Sulfur in aerosol, ranging in oxidation state from $S^{-II}$ to $S^{+IV}$, has also been identified in smaller quantities through the use of XANES techniques (Huggins et al., 2000;Eatough et al., 1978;Cao et al., 2015;Higashi and Takahashi, 2009;Craig et al., 1974;Cozzi et al., 2009). Primary sources of reduced sulfur to the

atmosphere are mainly attributed to gaseous emissions from volcanic gases, hot springs, bacteria, vehicle exhaust, and oil refineries (Cozzi et al., 2009;Andersson et al., 2006). While detected in the solid phase, reduced sulfur in aerosols has not been extensively studied. Reduced aerosol sulfur may reflect a contribution from primary emission sources or may result from condensation of volatile reduced sulfur gaseous phases, but the ultimate origin of reduced S in aerosol particulates remains unresolved.

Here we use S-NEXFS to investigate the composition and oxidation state of sulfur in ambient aerosols and common primary emission source samples. Previous S-NEXFS studies characterized a limited number of ambient aerosol samples in bulk, providing an average view of the sulfur oxidation state and composition in particulate matter





(Higashi and Takahashi, 2009;Takahashi et al., 2006;Pongpiachan et al., 2012a;Pongpiachan et al., 2012b;Long et al., 2014). Recent studies of aerosol iron and selenium have demonstrated that analyzing compositional differences between large particles (> 1 μm) and the bulk sample can help elucidate aerosol sources and transformations (Oakes et al., 2012b;De Santiago et al., 2014). Following a similar approach, we used a combination of bulk and individual

particle S-NEXFS analyses to characterize aerosol sulfur in diverse rural and urban samples and from different emission sources. The spatial distribution of sulfur oxidation states was also characterized in individual particles by mapping particles at different incident energies to gain further insights into the processes surrounding the formation of reduced sulfur species in aerosol.

## 2 Methods

**2.1 Ambient Aerosol Collection**

PM2.5 samples were collected from urban and rural locations in and around Atlanta, GA. The five sampling sites, South DeKalb, Fire Station 8, Jefferson Street, Yorkville, and Fort Yargo State Park, are associated with two ongoing studies, the SouthEastern Aerosol Research and Characterization (SEARCH) study (Edgerton et al., 2005;Edgerton et al., 2006;Hansen et al., 2003) and the Assessment of Spatial Aerosol and Composition in Atlanta

(Butler et al., 2003). Samples were collected on Zefluor filters over 24-h periods at a flow rate of 16.7 L min$^{-1}$ using PM2.5 cyclone inlet samplers (URG, Chapel Hill, NC). The multichannel particle samplers were mounted approximately 2.5 m off the ground. Three samples were collected from Yorkville (33°55'42.3″N, 85°2'43.68″W), a rural background site for the Atlanta area. Two samples were collected from Jefferson Street (33°46'38.94″N, 84°24'59.58″W), an urban collection site, which is not immediately influenced by any individual

source or roadside traffic. Five samples were collected from Fort Yargo State Park (33°58'4.18″N, 83°43'29.16″W), a rural setting that is frequently impacted by biomass burning plumes and power plant emissions. Four samples were collected in South DeKalb, a mixed commercial–residential area, approximately 1 km from a major interstate (33°41'16.48″N, 84°17'25.26″W). Eight samples were collected from Fire Station 8, located in an industrial area within close proximity to a rail yard, a fire station, and an intersection with heavy diesel truck traffic (33°48'6.01″N,

84°26'8.75″W; Table S1).

**2.2 Emission Source Collection**

PM2.5 samples were collected from gasoline and diesel exhaust, biomass burning, and coal fly ash (Oakes et al., 2012a). Emissions from ultralow sulfur diesel fuel running a 10.8 L engine and conventional gasoline fuel running a 3.3 L engine were collected according to US Environmental Protection Agency protocols under typical

urban driving conditions (Liu et al., 2008;Oakes et al., 2012a). Polydisperse coal fly ash, provided by The Southern Co., from an electrostatic precipitator of a midsized coal-fired power plant was aerosolized and collected with a cyclone inlet sampler to separate the PM2.5 fraction (Oakes et al., 2012a). Smoke produced from the burning of materials collected from coniferous and deciduous trees native to Georgia, USA, was sampled during a controlled biomass burning experiment using a PM2.5 cyclone inlet sampler placed 3.5 m above the burn area at a flow rate of

16.7 L min$^{-1}$ for approximately 30 min. The ash produced from the biomass burning experiment was analyzed in the





same manner. The above primary emission source samples were collected on polytetrafluoroethylene (Zefluor) filters (Longo et al., 2014). In addition, two commercially available bacteria samples, *Azotobacter vinelandii* (Sigma A2135) and *Bacillus subtilis* (Sigma B4006), were homogenized with agate mortar and pestle and mounted on a cellulose acetate filter for analysis.

All samples were immediately stored after preparation in clean Petri dishes at −20 °C until analysis. Preparation for synchrotron analyses consisted of mounting an approximate 0.5 cm × 0.5 cm section of ambient aerosol and primary emission source filters over a slot on an aluminum support.

### 2.3 Sulfate Standards

In order to create a database of inorganic sulfate standards necessary for data analysis described below, ten
compounds were analyzed using the same experimental setup and synchrotron system as the ambient aerosols. These standards included, ammonium sulfate (CAS 7783-20-2), barite, copper(II) sulfate (CAS 7758-98-7), gypsum, iron ammonium sulfate (CAS 7783-85-9), iron(III) sulfate (CAS 15244-10-7), jarosite ($KFe^{3+}_3(OH)_6(SO_4)_2$), magnesium sulfate (CAS 7487-88-9), potassium sulfate (CAS 7778-80-5), and sodium sulfate (CAS 7757-82-6).  All standards were homogenized with an agate mortar and pestle and mounted on a cellulose acetate filter for analysis. A database
of sulfate standards is provided in the Supporting Material.

### 2.4 Synchrotron-based Spectromicroscopy

Samples were analyzed on the X-ray fluorescence microscope located at beamline 2-ID-B at the Advanced Photon Source, Argonne National Laboratory. The beamline is optimized to examine samples over a 1–4 keV energy range using a focused X-ray beam with a spot size of approximately 60 $nm^2$ (McNulty et al., 2003)). Sulfur near-edge
X-ray fluorescence spectroscopy (S-NEXFS) data were collected in two modes that differ based on spatial resolution. In the first mode, individual sulfur-rich particles with a diameter of greater than 1 µm were identified in X-ray fluorescence maps; these particles were then interrogated with micro S-NEXFS. The individual sulfur-rich particles seen in X-ray fluorescence maps are obvious contributors to the total sample sulfur. However, much of the total sulfur on an aerosol filter with a PM2.5 inlet can also be contained in particles that are smaller than 1 µm or less sulfur-rich
and therefore less apparent in X-ray fluorescence maps. Therefore, in the second mode, large areas of the filters were also examined with an unfocused beam (spot size = 0.25 mm).

In order to maximize the number of samples analyzed in the allotted time, X-ray fluorescence maps were created for a subset of samples by rastering the focused beam in 0.5 µm steps with an incident energy of 2535 eV. At this resolution, individual sulfur-rich particles were clearly discernible. S-NEXFS spectra were scanned a 50 eV range
centered at 2485 eV in 0.33 eV steps, using a 1 s dwell time at each step. Each S-NEXFS measurements for both bulk and individual sulfur-rich particles were repeated at least 3 times, in a single location, creating a minimum effective dwell time of 3 s. X-ray spectromicroscopy data were collected using an energy dispersive silicon drift detector (Vortex with a 5 $mm^2$ sensitive area). A flow of helium was introduced between the X-ray optical hardware and the sample to reduce elastically scattered X-ray background. An in-line monitor stick coated with a zinc sulfate standard
was measured in parallel with each sample in order to identify and correct for any potential drift in monochrometer



energy calibration that can occur during analyses (de Jonge et al., 2010)). Clean areas of filter were examined as blanks and showed negligible background signal.

S-NEXFS provides essentially the same information as another commonly cited technique, S-XANES (sulfur X-ray absorption near-edge structure) spectroscopy. The two techniques differ primarily in the method of signal

detection. S-NEXFS uses the X-ray fluorescence signal, which is inversely proportional to the absorption signal used in a XANES measurement.

Additionally, XRF microscopy was conducted in two modes. First, to map the distribution of sulfur regardless of chemical state an incident energy above the S K-edge (2535 eV) was used, and samples were raster through the focused beam in 0.5 µm steps, as mentioned above. In the second mode, the spatial distribution of $S^0$ and $S^{+VI}$

oxidation states were quickly mapped by selecting an incident X-ray energy tuned to their specific whitelines at 2471 eV and 2480 eV respectively. XRF maps were completed in 0.3 µm spatial resolution and 0.5s dwell time per pixel. X-ray focusing was adjusted to maintain a consistent beam spot size at both energies, however some shift in the final image did occur. The images were aligned using the stack registration function of the Fiji software (Schindelin et al., 2012).

**2.5 Data Analysis**

S-NEXFS data were normalized to create a relative intensity value of approximately 1 for post edge area of the spectra. The data was also processed using a three-point smoothing algorithm built into the software package Athena to remove high frequency noise (Ravel and Newville, 2005). The relative contribution of each oxidation state can be determined by assuming the entire area under the whiteline peaks is representative of total sulfur, and the area

under each whiteline peak represents the relative contribution of each oxidation state (Huffman et al., 1991;Xia et al., 1998). The area under the whiteline peaks was determined using the Gaussian peak fitting function of Athena (Ravel and Newville, 2005). Because of the relatively low contribution of other sulfur oxidation states, only sulfur species containing $S^{+VI}$ oxidation state could be further characterized via linear combination fitting. Linear combination fitting is an effective tool for the deconvolution of spectra of known mixtures (Longo et al., 2014;Long et al., 2014;Huffman

et al., 1991;Solomon et al., 2003;Prietzel et al., 2011). Using Athena software, individual particle and bulk S-NEXFS spectra were fit with previously characterized sulfate standard materials using a linear combination approach to determine both speciation and relative abundance of sulfate phases (Ravel and Newville, 2005). Athena uses a nonlinear, least squares minimization approach to fit spectra of unknown materials with spectra of standard materials and computes an error term, *R* factor, to quantify the goodness of fit produced by a particular linear combination of

standard S-NEXFS spectra. The linear combination of standards that yielded the lowest *R* factor reflects the best fit (Ravel and Newville, 2005).

**3 Results**

**3.1 Oxidation State**

*Azotobacter vinelandii*, *Bacillus subtilis*, gasoline and diesel exhaust, biomass burning, and coal fly ash were

only characterized at the bulk level (approximately 0.28 mm² filter area). These common primary emission sources,





with the exception of one bacteria sample, contained sulfur solely in the $S^{+VI}$ oxidation state. *Azotobacter vinelandii* was the only emission source that contained both oxidized and reduced sulfur, with approximately 44% $S^{+VI}$ and 56% $S^0$ (Figure 1).

At the bulk level, the oxidation state of sulfur in ambient PM2.5 samples is dominated by $S^{+VI}$; only two samples from South DeKalb contain $S^0$, at quantities of less than 10% total sulfur (Figure 2). In contrast, individual particles with aerodynamic diameters of greater than 1 μm consistently contain both $S^{+VI}$ and $S^0$ (Figure 3). Only one out of twenty-three individual particles analyzed did not contain $S^0$ at some detectable level. At Fire Station 8 and Jefferson Street, the individual particles sampled contain on average 10% $S^0$. Individual particles from South DeKalb average only 4% $S^0$. The rural individual particles sampled from Fort Yargo and Yorkville contain on average 5% and 9% $S^0$, respectively. Multi-energy maps revealed that $S^{+VI}$ was present throughout the aerosol particle. Often the highest concentration of $S^{+VI}$ was in the center of the particle, where the most mass is present (Figure S1). The spatial distribution of $S^0$ was more varied. In some cases, the $S^0$ was concentrated in the center of the particle with a less prominent ring on the outside of the particle. In other cases, the $S^0$ was most concentrated in only one area of the particle (Figure S1).

## 3.2 Sulfate Composition

The relative contribution of different sulfur species can be determined by the deconvolution of S-NEXFS spectra with linear combination fitting (Prietzel et al., 2011;Ravel and Newville, 2005). In these samples, only $S^{+VI}$ could be further characterized through linear combination fitting because it was by far the most abundant species of sulfur present in the samples. The regions of the S-NEXFS spectra represented by the lower sulfur oxidation states did not possess sufficient detail to yield compositional information. Because of spectral similarities between various metal sulfates, linear combination fits using copper(II) sulfate or iron(III) sulfate often yielded fits with similar *R* factors. This makes the absolute determination of metal sulfate composition difficult, thus, iron(III) sulfate, copper(II) sulfate, and jarosite are referred to collectively as metal sulfates. The specific compositional information derived from the best linear combination fits, i.e. iron(III) sulfate, copper(II) sulfate, or jarosite, is presented in Table S2 and S3, and suggests that a combination of iron and copper sulfates are likely present in these samples. In common primary emission sources and ambient PM2.5, $S^{+VI}$ corresponded to sulfate aerosol. In emission sources, only biomass burning, coal fly ash, and diesel exhaust had robust enough signals to characterize specific sulfate composition, and each source had a unique sulfate composition (Table S3). Biomass burning contained potassium sulfate ($100 \pm 0.005\%$). Coal fly ash contained gypsum ($100 \pm 0\%$), the mineral form of calcium sulfate. Diesel exhaust contains ammonium sulfate ($70 \pm 11\%$) and metal sulfate ($30 \pm 12\%$).

The composition of sulfate was generally consistent within a sampling location (Table S3 and Figure 4). At the bulk level, rural sampling locations contain a mix of ammonium sulfate and metal sulfate. Yorkville samples are comprised of $61 \pm 9\%$ ammonium sulfate and $39 \pm 9\%$ metal sulfate, and Fort Yargo samples contain $84 \pm 7\%$ ammonium sulfate and $16 \pm 7\%$ metal sulfate. Bulk urban samples all contain a large fraction of ammonium sulfate. Samples from Fire Station 8 contain ammonium sulfate ($84 \pm 7\%$) mixed with metal sulfate ($14 \pm 16\%$), and Jefferson Street contains ammonium sulfate ($51 \pm 8\%$) combined with metal sulfate ($50 \pm 10\%$). South DeKalb has three different compositional groups at bulk level: (1) ammonium sulfate ($63 \pm 7\%$), metal sulfate ($29 \pm 6\%$), and potassium





sulfate (8 ± 13%); (2) ammonium sulfate (42 ± 14%) and metal sulfate (58 ± 14%); and (3) gypsum (43 ± 15%) and metal sulfate (58 ± 15%).

Only individual particle ($1 < D_p < 2.5$ µm) S-NEXFS spectra from Fire Station 8, Fort Yargo, and South DeKalb contained enough sulfur to determine the sulfate composition (Table S3 and Figure 4). In total, 17 individual
particle S-NEXFS spectra were used to gain insights on the sulfate composition of ambient PM2.5. Consistent with the bulk results, individual particles sampled from Fire Station 8 are largely ammonium sulfate and metal sulfate, with the remainder made of gypsum. Individual particles from South DeKalb contain either ammonium sulfate or metal sulfate. Individual particles from Fort Yargo contain a mixture of mostly ammonium sulfate and potassium sulfate with additional contributions from metal sulfate and gypsum. Spectral signals of samples from Jefferson Street and
Yorkville were not strong enough to characterize the composition of sulfate in the ambient PM2.5 samples.

**4.0 Discussion**

Ammonium sulfate and gypsum are commonly identified phases in studies of aerosol sulfate composition (Long et al., 2014;Higashi and Takahashi, 2009;Takahashi et al., 2006). Consistent with these studies, ammonium sulfate was found across all urban and rural sampling locations, and either ammonium sulfate or gypsum were a major
constituent of all bulk samples. At the individual particle level, ammonium sulfate was also a primary contributor to sulfate composition; however, potassium and metal sulfates became more apparent. Our results showed that the sulfate in biomass burning was dominated by potassium sulfate, which suggests biomass burning may be the primary source of the potassium sulfate in ambient aerosol samples. Potassium is often used as a tracer for biomass burning in source apportionment studies (Viana et al., 2008), which show that biomass burning can contribute up to 40% of total PM2.5
in Georgia's spring months and 10% of total PM2.5 in Georgia's summer months (Tian et al., 2009). Organosulfates have recently been identified as a significant component of sulfate aerosol (Liao et al., 2015;Schmitt-Kopplin et al., 2010;Surratt et al., 2008;Xu et al., 2015). In this study, organosulfates were not identified as a constituent of sulfate aerosol;  however, the concentrations organosulfates are expected to occur at in ambient aerosol could be below the detection limit of NEXFS techniques (Longo et al., 2014;Oakes et al., 2012a;Oakes et al., 2012b).

Metal sulfates were identified as a constituent of all the bulk ambient particulate matter samples and 53% of individual particle spectra. Linear combination fitting of S-NEXFS spectra suggests that iron(III) sulfates are an important group of metal sulfates in ambient Atlanta aerosol (Table S2 & S3). Furthermore, an Fe-XANES study of ambient Atlanta particulate matter identified iron(III) sulfates, which accounted for approximately 20% of the total iron (Oakes et al., 2012a). Iron sulfate represent a very soluble, bioavailable, form of iron that has been found in
ambient aerosol (Zhang et al., 2014;Moffet et al., 2012;Schroth et al., 2009;Oakes et al., 2012a). Diesel exhaust was the only primary emission source to contain metal sulfates in this study, and because source apportionment studies show that diesel emissions can account for approximately 5.3% of PM2.5 in Atlanta (Hu et al., 2014), it is unlikely that diesel exhaust alone can explain the quantities of metal sulfates found in our ambient aerosol samples. Iron sulfate may also be the end product of acidic reactions hypothesized to occur in the atmosphere that are responsible for
solubilizing iron. Correlations between the sulfur content and iron solubility have previously been used to suggest this mechanism plays a role in shaping the composition of iron in ambient Atlanta aerosol (Oakes et al., 2012a). Briefly, sulfuric acid solubilizes more crystalline iron phases resulting in a solution rich in soluble iron and sulfate that can





become internally mixed and potentially precipitate out as iron(III) sulfate (Moffet et al., 2012;Zhang et al., 2014;Oakes et al., 2012a). Evidence of similar processes has been found for copper (Fang et al., 2015), which is another metal that is likely in these samples (Table S2 and Table S3). The metal sulfates seen in ambient Atlanta aerosol are likely the product of both primary emission sources as well as secondary acidic reactions.

In the multi-energy maps, $S^0$ always occurs with $S^{+VI}$ (Figure S4), suggesting they are either co-emitted as a primary source or linked by a secondary formation process that occurs in the atmosphere. Up to 20% of the sulfur in individual particles was in reduced form, as $S^0$. Previous examinations of ambient aerosol have reported less than 5% of total sulfur as reduced sulfur (Cozzi et al., 2009;Long et al., 2014) and have suggested incomplete combustion or incinerator emissions as the likely source (Andersson et al., 2006;Bao et al., 2009;Matsumoto et al., 2006). Common

primary emission sources, such as gasoline and diesel exhaust, coal fly ash, and biomass burning, did not contain $S^0$ as a readily identifiable constituent in bulk S-NEXFS spectra, leaving the source of reduced S in ambient Atlanta aerosol unresolved. Here, bulk S-NEXFS of *Azotobacter vinelandii* revealed this bacterium to be the only analyzed potential emission source to be enriched in $S^0$. Microbial cells are increasingly recognized as an important natural component of aerosol (Burrows et al., 2009;Bauer et al., 2002). The ubiquitous distribution of bacteria makes

aerosolized soil bacteria, such as *Azotobacter vinelandii*, another potential primary source of $S^0$. Furthermore, $S^0$ was commonly found in individual particles, which constitute the largest particle size fraction of these samples (>1 μm). This could further support the hypothesis that the $S^0$ is from aerosolized soil bacteria, which would reside in larger particles.

The reduced sulfur found in ambient aerosol particles could also be a result of secondary formation processes

that occur in the atmosphere. In a previous study, $S^{+IV}$ found in ambient particle matter was attributed to the absorption and incorporation of sulfur dioxide ($S^{+IV}$) onto atmospheric particulate matter (Higashi and Takahashi, 2009). Theoretically, a similar mechanism could help explain the finding of $S^0$ in ambient aerosols, however, gaseous phases of reduced sulfur compounds are unlikely to absorb onto an aerosol surface or condense without undergoing oxidation (Alexander et al., 2005;Liao et al., 2003). Furthermore, reduced sulfur species on the surface of a particle would be

easily oxidized by ozone, oxygen, or the hydroxyl radical, suggesting that only reduced sulfur inside of a heterogeneous particle would be likely to survive (Long et al., 2014), suggesting that the $S^0$ likely has a primary source.

The composition and oxidation state of sulfur in ambient aerosol provides insights for the atmospheric chemistry involving sulfur as well as metals. More than 25% of the bulk sulfate composition can be attributed to metal sulfates, which cannot be accounted for by primary sources alone (Hu et al., 2014). The solubilization of metals with

sulfuric acid during atmospheric transport likely plays a role in the forming the metal sulfates observed in this study as well as others (Oakes et al., 2012a). Reduced sulfur was also found to account for up to 20% of the total sulfur in individual particles, which is a higher fraction than typically observed in ambient aerosol samples (Long et al., 2014;Cozzi et al., 2009).  As is the case with previous studies that have noted reduced sulfur in ambient aerosol samples, the $S^0$ is likely from a primary emission source. Incomplete combustion is the most commonly cited source

of reduced sulfur compounds found in aerosol (Long et al., 2014;Cozzi et al., 2009;Bao et al., 2009;Matsumoto et al., 2006;Andersson et al., 2006); however, in this study, a bacterium was the only potential primary emission source to



contain $S^0$ at the bulk level. This suggests that aerosolized bacteria may contribute to the $S^0$ seen in ambient Atlanta aerosol.





**Acknowledgments.** This material is based upon work supported by the National Science Foundation under Grant OCE-1357375. The data used to produce these results is available upon request to the corresponding author. Any opinions, findings, and conclusions or recommendations expressed in this material are those of the authors and do not necessarily reflect the views of the National Science Foundation. Use of the Advanced Photon Source is
5   supported under the U.S. Department of Energy contract No. DE-AC02-06CH11357.

**Supporting Information.** This article is accompanied by additional tables and figures available in the Supporting Information.





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





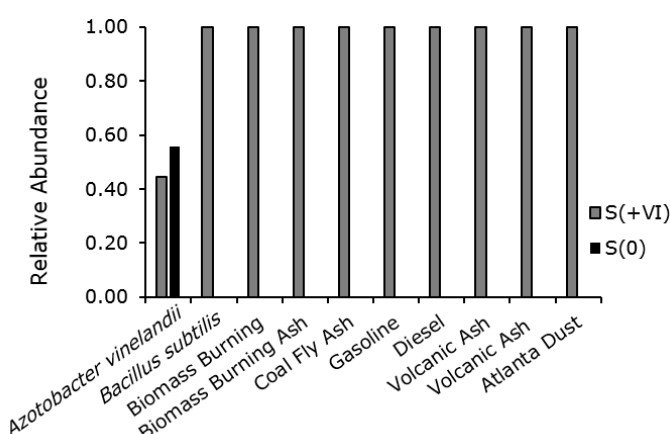

**Figure 1: Oxidation state of sulfur in common emission sources** Common emission sources were generally dominated by $S^{+VI}$ (grey). Only the bacteria sample *Azotobacter vinelandii* showed a signal for $S^0$ as well as $S^{+VI}$.


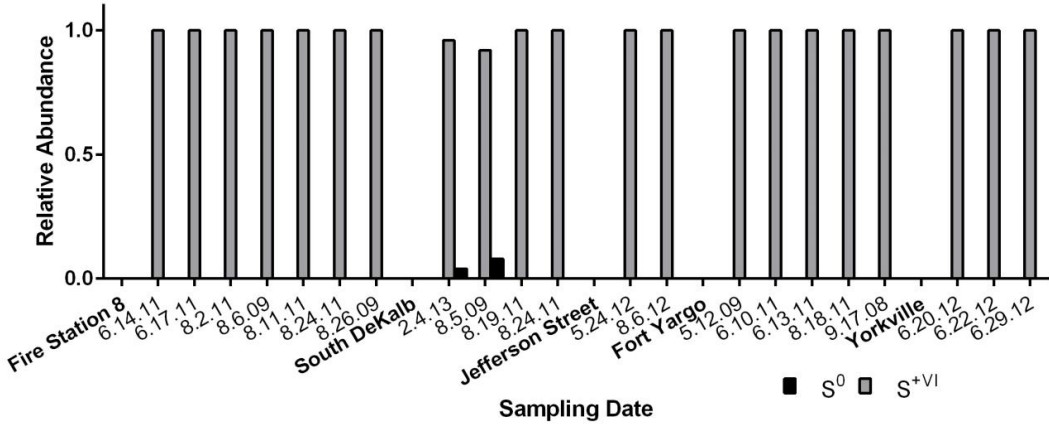

**Figure 2: Bulk oxidation state of ambient particulate matter samples** Ambient particulate matter samples were characterized at the bulk level for oxidation state. Here, the fractional relative abundance of $S^0$ (black) and $S^{+VI}$ (grey) are shown for each of the ambient particulate matter samples. $S^0$ is only present in two samples collected from South DeKalb. The remaining samples contain only $S^{+VI}$ when examined in bulk.





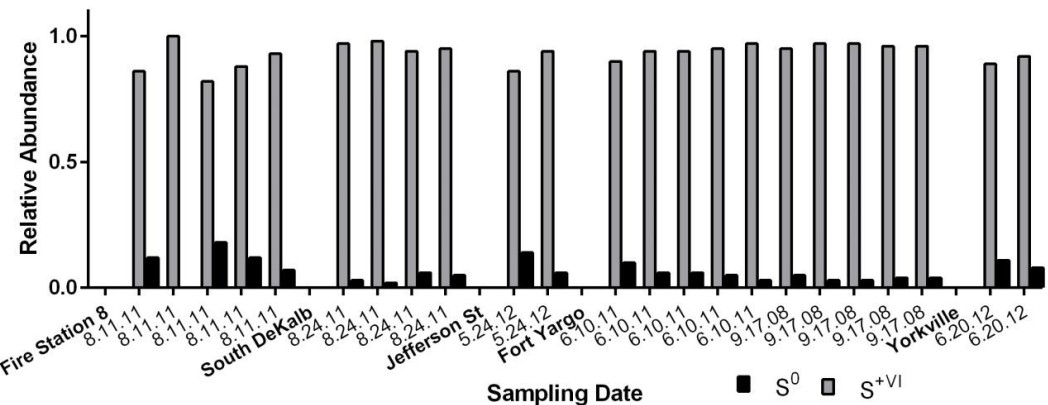

**Figure 3: Oxidation state of individual aerosol particles** The oxidation state of sulfur was examined in individual particles from a subset of samples. The fractional relative abundance of $S^0$ (black) and $S^{+VI}$ (grey) is shown for each particle interrogated with S-NEXFS. At the individual particle level, sulfur is consistently seen at both the $S^0$ and $S^{+VI}$ oxidation states.



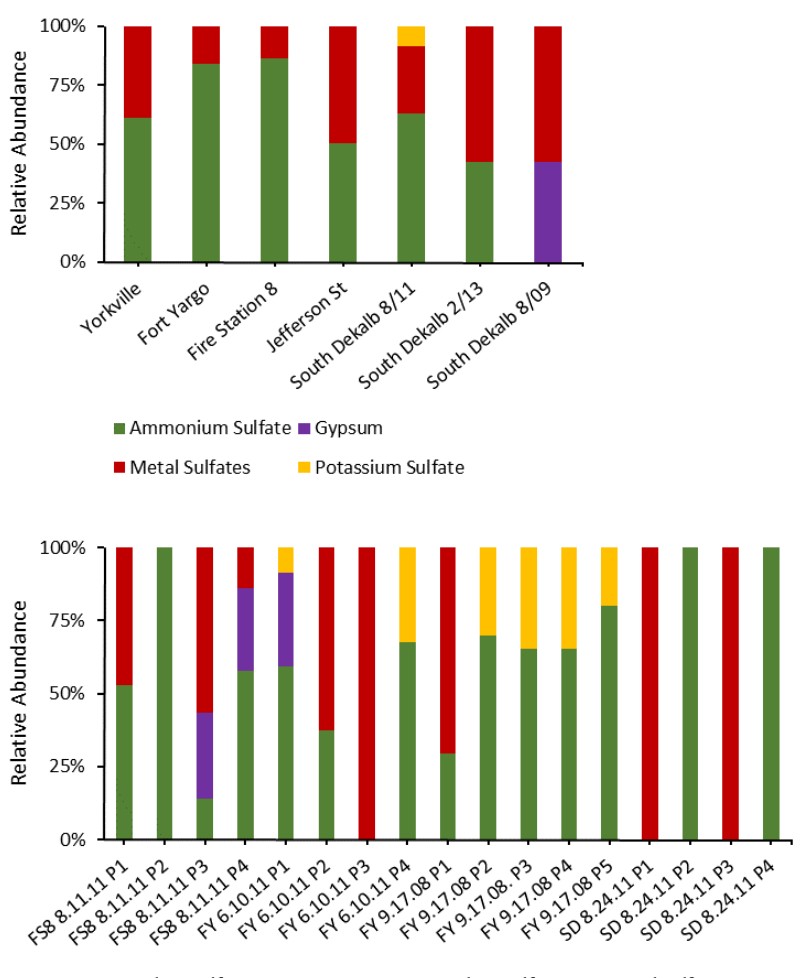

**Figure 4: Composition of aerosol sulfate** Bulk (top) and individual particle (bottom) composition of sulfur in ambient particulate matter. For bulk samples, the composition of sulfate was consistent for all samples within a particular sampling location, with the exception of South DeKalb. These samples had three different compositional groups, and the month and year of collection are provided next to the samples. For individual particles, the sulfate composition was able to be determined for samples from Fire Station 8 (FS8), Fort Yargo (FY), and South DeKalb (SD). Different particles from the same samples are numbered sequentially (P1-P5). Individual particles analyses include particles with nominal diameters between 1 and 2 µm.