# Peer review of "Composition and oxidation state of sulfur in atmospheric particulate matter"

_Atmospheric Chemistry and Physics, 2016_

## Referee Comment (RC1) · Anonymous Referee #1 · 24 Jun 2016

This paper discusses the compositon and the oxidation state of sulfur in atmospheric aerosols collected in the general Atlanta area. The overall work is of interest to the commiunity and is well written and has adequate discussion regarding the basic results. Some better presentation of the data and results and clarification of data processing methods however, should be included in at least the supplemental materials.

Specific Comments:

page 5, line 15: I very much like the addition of the sulfate standard database in the supplemental material. However, can the authors address the potential of self-absorption effects of sulfate standards? Particularly as this may possibly contribute to broadness and amplitude reduction of some of the standard peaks? Can the authors elaborate more on wether these data were collected in the bulk or microscale mode?
Any more info on the standards particle size, more than "homogenized"?

Suplemental material should at least include some plots of NEXFS spectra and their fit results of some typical sample to be able to evaluate the data quality and method of fitting. Both types of fitting with the linear combination of sulfate standards and the use of gaussian functions should be shown.

page 5, line 34: The authors follow a rigourous methods of following the monochromator energy drift. However, they do not reference what was mono energy was calibrated with initially? This information is critical to allow comparison to any other published dataset of sulfur spectroscopy. Was there any calibration of the sulfur concentration in the samples? If so, was this a theoretical calculation or an empirical calibration. It would be useful to discuss not only the relative changes in oxidation states, but the overall concentrations observed as well. This information would help in examining the various sources.

Fig S1. I would like to see more information regarding the multiple energy maps. What are the units of the map for each sulfur oxidation state? How were the units determined. Is the intensity here the intensity of the sulfur at each of the white line energies, or was a fit done to calculate the various proportions of each oxidation state? The latter would be a much more rigourous method, as there could be intensity of S0 and SVI is not completely unique to each of the white lines – that is there are contributions of each species to each of the measured energies. A proper method would be to measure the intensities at least N+1 energies (given S0 and SVI, N=2) and do a linear combination fit to determine the N species present at each map pixel. The choice of white line energies comared to the standard library seems off – there white line in the library is at $\sim$2483 eV for sulfate typically, whereas the maps were stated to have been measured at 2480 eV. One would expected, based on the apparent energy calibration, that S0 would appear at 2474 eV. Given the 2480 energy, one may observe some signifcant intensity of sulfates such as the ammonium sulfates around the energies of S0. Could this mislead the interpretation of S0 in the multiple energy maps if a careful fitting was

not performed?

Technical Corrections : page 5, line33: Did the experiment actually use Vortex SDD with 5mm2 area? This seems very small for standard XRF analysis. The more standard value for analysis is 50mm2. page 9, line 5: Text refers to Figure S4. This does not exist. Do the authors mean Figure S1?

---

## Author Comment (AC1) · 6 Jul 2016

We thank the reviewer for his/her very helpful comments and suggestions.

Reviewer 1 voiced concern about the potential effects of self-absorption for the sulfate standards, and requested further detail on the preparation of these standards. Sulfur standards were ground using an agate mortar and pestle to the consistency of a fine talcum powder (approximately 10 microns). A cellulose acetate filter was gently dredged through a small quantity (less than 1 mg) of powder placed on a microscope slide. This procedure produced a thin and almost imperceptible coating on the filter in order to limit the thickness and thus self-absorption. S-NEXFS spectra of sulfate standards were collected in bulk mode. Self-absorption must be carefully controlled when measuring fluorescent X-rays from thick specimens; however, the effects of selfabsorption are limited to the region of the spectrum above the K-edge (Iida and Noma, 1993; Bajt et al., 1993). In our repeated measurements, the post-edge features were consistent and reproducible which allows us to distinguish between sulfate standards.

The energy was calibrated using an elemental sulfur standard (S0) measured at beamline 2-ID-B. The whiteline energy of the elemental sulfur standard was aligned to 2472 eV (Cozzi et al., 2009). All subsequent data uses 2472 eV as the reference energy for S0 during the data alignment mentioned by Reviewer 1. Furthermore, for every measurement two spectra are collected: the specimen and an aluminum sulfate standard on the monitor stick. This approach means that all spectra are referenced to the aluminum sulfate standard and the initial calibration is not as crucial on this beamline.

Reviewer 1 also wanted more detailed information to assist with the interpretation of the multi-energy maps presented in Figure S1. For the multi-energy maps, the units are raw counts from the detector. At this beamline, units such as mass per area are not provided. A more intense signal, that is the more counts present, indicates greater concentrations of sulfur in that region of the sample. The maps were generated at the whiteline energies of S0 and S+VI. The energies referenced in the methods are the actual settings used during experiment; in other words, these energies are not calibrated with the monitor stick. To identify the correct whiteline energies for the multi-energy mapping, a S-NEXFS spectrum was collected for the particle of interest immediately before mapping. The corresponding whiteline energies for S0 and S+VI were then taken directly off this spectrum. Individual maps were then collected at the whiteline energies determined for S0 and S+VI. Although it is possible that the energy drifted during mapping, the interval between the two measurements is short enough that this is not a problem. Due to the 6 ev difference in the whiteline energies of the S0 and S+VI oxidation states, we do not expect significant overlap between the two oxidation states in multi-energy mapping. Energy drift during mapping may reduce signal intensities of the S0 and S+VI oxidation states but the overall distribution patterns of the sulfur oxidation states in an individual particle should remain relatively unaffected.

Fitting was not done to determine the relative abundance of sulfur at each oxidation state for each pixel. Instead, more accurate data from individual particle spectroscopy was collected and presented in the paper and supplement.

The reviewer also suggested that S-NEXFS data be provided in the supplement; therefore, additional figures will be added to the supplement that show exemplars of both the linear combination fitting and Gaussian peak fitting that were used to determine the sulfate composition and relative abundance of each oxidation state, respectively. These synchrotron-based measurements form the basis of this study. The concentrations of sulfur were not measured in this study.

All of the above information will all be added to the manuscript and/or supplement, should the paper be accepted.

We thank the reviewer for noting a few technical corrections as well. The reviewer is correct that the area of the Vortex SSD was 50 mm2, and this will be corrected in the manuscript. Also, text referring to Figure S4 will be corrected to Figure S1.

---

## Referee Comment (RC2) · Anonymous Referee #2 · 11 Jul 2016

General comments:

This paper presents the results of a study on the speciation of sulfur in ambient aerosol samples from the greater Atlanta area. The topic is of interest to this journal and studied with an appropriate technique. The article is generally well written, but the description of data collection and processing needs to be improved before publication.

Specific comments:

Page 3, line 27: Given that secondary sources are discussed for reduced sulfur and the mention of organosulfates in the Discussion section, I think a short sentence on secondary S(+VI) would be warranted.

Page 5, line 13: I commend the inclusion of the standards database in the supplementary information. I do however think that more information on the nature of the measured standards (e.g. particle size) would vastly improve the usefulness of this database.

Page 5, line 29: the description of the settings doesn't add up properly to me. It states that a 50 eV range was scanned in 0.33 eV steps with a dwell time of 1 s per step. If three full spectra were collected per particle or area (which is how I'd interpret the following sentence), how can the total dwell time be only 3 s? Also, given that multiple spectra were collected for each particle, did you observe any indication of beam damage (especially for S(0))?

Page 5, line 34: It is stated how a potential drift in energy calibration could be monitored. However, no mention is made of the original method of energy calibration. Was a sulfur standard used and if so, which one?

Page 6, line 31: Some spectra and the associated fits should be shown so that readers are able to evaluate data and fit quality (if nor here, then at least in the Supplementary Information).

Page 6, line 34: The primary emission samples were only characterized at the bulk level. Given that most of the S(0) in the ambient samples was observed only for individual particles, wouldn't this lead to wrong conclusions regarding their comparability?

Page 8, line 22: This sounds as if this study comes to the conclusion that organosulfates were not present. However, as far as I can see no organosulfate standard was measured, nor are there any comparisons to literature data, so how do the authors come to this conclusion?

Table S1: It is clear from this table that measurements of individual particles were only taken for few of the collected samples, in particular at Fire Station 8. Is there any specific reason for that?

More information on the generation of the sulfur maps is needed, either in the Data

[Figure]

Analysis section or at least in the caption of figure S1. For example: are the maps normalized? What is the unit of the scale bars?

Technical corrections:

page 5, line 35: "monochrometer" should be "monochromator"

page 6, line 8: "raster" should be "rastered "
* * *

---

## Author Comment (AC2) · 22 Jul 2016

We thank this reviewer for their very helpful comments and suggestions.

Reviewer 2 sought further clarifications on the synchrotron-based methods used in this manuscript. Dwell time is used to refer to the time spent at each step in the S-NEXFS spectrum. In this case, because three spectra with a 1 s dwell time each were often necessary to get enough signal for a high quality spectrum, the effective dwell time was at least 3 s because a minimum of 3 s were spent at each energy step in the spectrum. With the 50 eV range and 0.33 eV step-size used in this study, measurements were taken at 150 energies per scan. For 150 energy steps with an effective dwell time of 3 s, the total time required to generate a S-NEXFS spectrum was approximately 7.5 minutes. We did oxidation tests on samples where we measured the same particle

repeatedly, creating effective dwell times of more than 10 s. Even with this longer dwell time per energy step, we saw no noticeable shift in oxidation state or the relative abundance of the oxidation states.

The energy was calibrated using an elemental sulfur standard (S0) measured at beamline 2-ID-B. The whiteline energy of the elemental sulfur standard was aligned to 2472 eV (Cozzi et al., 2009). All subsequent data uses 2472 eV as the reference energy for S0 during the data alignment mentioned by both reviewers. Furthermore, for every measurement two spectra are collected: the specimen and an aluminum sulfate standard on the monitor stick. This approach means that all spectra are referenced to the aluminum sulfate standard and the initial calibration is not as crucial on this beamline.

Sulfur standards were ground using an agate mortar and pestle to the consistency of a fine talcum powder (approximately 10 microns). A cellulose acetate filter was gently dredged through a small quantity (less than 1 mg) of powder placed on a microscope slide. This procedure produced a thin and almost imperceptible coating on the filter in order to limit the thickness and thus self-absorption.

Reviewer 2 also wanted more detailed information to assist with the interpretation of the multi-energy maps presented in Figure S1. For the multi-energy maps, the units are raw counts from the detector. At this beamline, units such as mass per area are not provided. A more intense signal, that is the more counts present, indicates greater concentrations of sulfur in that region of the sample. The maps were generated at the whiteline energies of S0 and S+VI. The energies referenced in the methods are the actual settings used during experiment; in other words, these energies are not calibrated with the monitor stick. To identify the correct whiteline energies for the multi-energy mapping, a S-NEXFS spectrum was collected for the particle of interest immediately before mapping. The corresponding whiteline energies for S0 and S+VI were then taken directly off this spectrum. Individual maps were then collected at the whiteline energies determined for S0 and S+VI. Although it is possible that the energy drifted during mapping, the interval between the two measurements is short enough that this

is not a problem. Due to the 6 ev difference in the whiteline energies of the S0 and S+VI oxidation states, we do not expect significant overlap between the two oxidation states in multi-energy mapping. Energy drift during mapping may reduce signal intensities of the S0 and S+VI oxidation states but the overall distribution patterns of the sulfur oxidation states in an individual particle should remain relatively unaffected.

Fitting was not done to determine the relative abundance of sulfur at each oxidation state for each pixel. Instead, more accurate data from individual particle spectroscopy was collected and presented in the paper and supplement. As the reviewer noted, individual particle spectroscopy was not collected for every sample in our collection. Individual particles were examined on samples other than those noted in Table S1; however, the particles did not generate useable spectra. Table S1 represents the number of spectra that were collected that could be used for further analysis.

Reviewer 2 also wanted clarification on the use of bulk emission source data. Emission sources were only characterized at the bulk level, and this poses a problem for data interpretation. We still use the emission source data for comparison with bulk ambient aerosol data and to a limited extent the individual particle data to glean any possible insights; however, we will further clarify that emission sources were collected in the bulk mode and any comparisons between the bulk emission source data and the ambient aerosol individual particle data are speculative.

In this manuscript, we postulate that organosulfates are not a major component of sulfate aerosol. We were able to account for the sulfate using our present database, which suggests that organosulfates likely do not account for a significant portion of the sulfate aerosol. We do, however, state that contributions near or below 10% could be below the detection limit of this method. Since this is approximately the level that organosulfates are hypothesized to occur, it makes sense that we are able to account for all sulfate present in the sample with our database, even if some small portion of organosulfate is present.

A short sentence will be added to the Discussion about the possibility of secondary formation of sulfates. Using just these techniques, it is unclear what the role of secondary sulfates are in these ambient samples; however, secondary formation processes could certainly contribute to the S(+VI) pool.

Figures will be added to the supplement (see attachments) that show exemplars of both the linear combination fitting and Gaussian peak fitting that were used to determine the sulfate composition and relative abundance of each oxidation state, respectively.

We thank the reviewer for noting a few technical corrections as well. These and the above comments will be addressed in the revised manuscript, should the paper be accepted.
* * *
[Figure]

[Figure]

Figure S2: Linear combination fits of representative bulk samples. The first derivative
was used to fit ambient aerosol samples (solid lines) with a database of sulfate standards.
The resulting linear combination fits are shown with open circles.

[Figure]

Figure S3: Gaussian peak fitting was used to determine the relative abundance of the S(0) and S(+VI) oxidation state in the aerosol samples. Athena's peak fitting protocol was used to fit Gaussian curves (dotted line) to the S-NEXFS spectrum (solid line) and to determine the area under the Gaussian curves.

[Figure]

---

## Editor Comment (EC1) · Hang Su (Editor) · 2 Oct 2016

Author Response:

We thank these reviewers for their very helpful comments and suggestions.

Comments from Referee #1

This paper discusses the composition and the oxidation state of sulfur in atmospheric aerosols collected in the general Atlanta area. The overall work is of interest to the community and is well written and has adequate discussion regarding the basic results. Some better presentation of the data and results and clarification of data processing methods however, should be included in at least the supplemental materials.

Specific Comments:

page 5, line 15: I very much like the addition of the sulfate standard database in the supplemental material. However, can the authors address the potential of self-absorption effects of sulfate standards? Particularly as this may possibly contribute to broadness and amplitude reduction of some of the standard peaks? Can the authors elaborate more on whether these data were collected in the bulk or microscale mode? Any more info on the standards particle size, more than "homogenized"?

Sulfur standards were ground using an agate mortar and pestle to the consistency of a fine talcum powder (approximately 10 microns). A cellulose acetate filter was gently dredged through a small quantity (less than 1 mg) of powder placed on a microscope slide. This procedure produced a thin and almost imperceptible coating on the filter in order to limit the thickness and thus self-absorption. S-NEXFS spectra of sulfate standards were collected in bulk mode. Self-absorption must be carefully controlled when measuring fluorescent X-rays from thick specimens; however, the effects of self-absorption are limited to the region of the spectrum above the K-edge (Iida and Noma, 1993; Bajt et al., 1993). In our repeated measurements, the post-edge features were consistent and reproducible which allows us to distinguish between sulfate standards.

This text was added to the Methods section under Sulfate Standards (page 5).

Supplemental material should at least include some plots of NEXFS spectra and their fit results of some typical sample to be able to evaluate the data quality and method of fitting. Both types of fitting with the linear combination of sulfate standards and the use of Gaussian functions should be shown.

Additional figures were added to the supplement that show exemplars of both the linear combination fitting and Gaussian peak fitting that were used to determine the sulfate composition and relative abundance of each oxidation state, respectively.

Figures S1 and S2 were added to pages 4 & 5 of the Supplementary Information.

page 5, line 34: The authors follow a rigorous method of following the monochromator energy drift. However, they do not reference what was mono energy was calibrated with initially? This information is critical to allow comparison to any other published dataset of sulfur spectroscopy.

Was there any calibration of the sulfur concentration in the samples? If so, was this a theoretical calculation or an empirical calibration. It would be useful to discuss not only the relative changes in oxidation states, but the overall concentrations observed as well. This information would help in examining the various sources.

Fig S1. I would like to see more information regarding the multiple energy maps. What are the units of the map for each sulfur oxidation state? How were the units determined? Is the intensity here the intensity of the sulfur at each of the white line energies, or was a fit done to calculate the various proportions of each oxidation state? The latter would be a much more rigorous method, as there could be intensity of S0 and SVI is not completely unique to each of the white lines – that is there are contributions of each species to each of the measured energies. A proper method would be to measure the intensities at least N+1 energies (given S0 and SVI, N=2) and do a linear combination fit to determine the N species present at each map pixel. The choice of white line energies compared to the standard library seems off – there white line in the library is at ~2483 eV for sulfate typically, whereas the maps were stated to have been measured at 2480 eV. One would expect, based on the apparent energy calibration, that S0 would appear at 2474 eV. Given the 2480 energy, one may observe some significant intensity of sulfates such as the ammonium sulfates around the energies of S0. Could this mislead the interpretation of S0 in the multiple energy maps if a careful fitting was not performed?

whiteline energies of the $S^0$ and $S^{+VI}$ oxidation states, we do not expect significant overlap between the two oxidation states in multi-energy mapping. Energy drift during mapping may reduce signal intensities of the $S^0$ and $S^{+VI}$ oxidation states but the overall distribution patterns of the sulfur oxidation states in an individual particle should remain relatively unaffected.

Fitting was not done to determine the relative abundance of sulfur at each oxidation state for each pixel. Instead, more accurate data from individual particle spectroscopy was collected and presented in the paper and supplement.

This text was added to the Methods Section under Synchrotron-based Spectromicroscopy (page 6).

Technical Corrections:

page 5, line33: Did the experiment actually use Vortex SDD with 5 mm$^2$ area? This seems very small for standard XRF analysis. The more standard value for analysis is 50 mm$^2$.

The reviewer is correct that the area of the Vortex SSD was 50 mm$^2$ (page 6).

page 9, line 5: Text refers to Figure S4. This does not exist. Do the authors mean Figure S1?

Text referring to Figure S4 was corrected (page 9).

Comments from Referee #2

General comments:

This paper presents the results of a study on the speciation of sulfur in ambient aerosol samples from the greater Atlanta area. The topic is of interest to this journal and studied with an appropriate technique. The article is generally well written, but the description of data collection and processing needs to be improved before publication.

Specific comments:

Page 3, line 27: Given that secondary sources are discussed for reduced sulfur and the mention of organosulfates in the Discussion section, I think a short sentence on secondary S(+VI) would be warranted.

  A short sentence was added to the Discussion about the possibility of secondary formation of sulfates. Using just these techniques, it is unclear what the role of secondary sulfates are in these ambient samples; however, the high concentrations of metal sulfates suggests that secondary processes do influence the composition of S(+VI) in ambient aerosols.

This text was added to the Discussion (page 9).

Page 5, line 13: I commend the inclusion of the standards database in the supplementary information. I do however think that more information on the nature of the measured standards (e.g. particle size) would vastly improve the usefulness of this database.

  Sulfur standards were ground using an agate mortar and pestle to the consistency of a fine talcum powder (approximately 10 microns). A cellulose acetate filter was gently dredged through a small quantity (less than 1 mg) of powder placed on a microscope slide. This procedure produced a thin and almost imperceptible coating on the filter in order to limit the thickness and thus self-absorption.

This text has been added to the manuscript in the Methods section under Sulfate Standards (page 5).

Page 5, line 29: the description of the settings doesn't add up properly to me. It states that a 50 eV range was scanned in 0.33 eV steps with a dwell time of 1 s per step. If three full spectra were collected per particle or area (which is how I'd interpret the following sentence), how can the total dwell time be only 3 s? Also, given that multiple spectra were collected for each particle, did you observe any indication of beam damage (especially for S(0))?

  Dwell time is used to refer to the time spent at each step in the S-NEXFS spectrum. In this case, because three spectra with a 1 s dwell time each were often necessary to get enough signal for a high quality spectrum, the effective dwell time was at least 3 s because a minimum of 3 s were spent at each energy step in the spectrum. With the 50 eV range and 0.33 eV step-size used in this study, measurements were taken at 150 energies per scan. For 150 energy steps with an effective dwell time of 3 s, the total time required to generate a S-NEXFS spectrum was approximately 7.5 minutes. We did oxidation tests on samples where we measured the same

particle repeatedly, creating effective dwell times of more than 10 s. Even with this longer dwell time per energy step, we saw no noticeable shift in oxidation state or the relative abundance of the oxidation states.

Dwell time was defined in the manuscript in the Methods under Synchrotron-based Spectromicroscopy (page 5-6).

Page 5, line 34: It is stated how a potential drift in energy calibration could be monitored. However, no mention is made of the original method of energy calibration. Was a sulfur standard used and if so, which one?

The energy was calibrated using an elemental sulfur standard (S0) measured at beamline 2-ID-B. The whiteline energy of the elemental sulfur standard was aligned to 2472 eV (Cozzi et al., 2009). All subsequent data uses 2472 eV as the reference energy for S0 during the data alignment mentioned by both reviewers. Furthermore, for every measurement two spectra are collected: the specimen and an aluminum sulfate standard on the monitor stick. This approach means that all spectra are referenced to the aluminum sulfate standard and the initial calibration is not as crucial on this beamline.

This text was added to the Methods section under Synchrotron-based Spectromicroscopy (page 5-6).

Page 6, line 31: Some spectra and the associated fits should be shown so that readers are able to evaluate data and fit quality (if nor here, then at least in the Supplementary Information).

Figures were added to the supplement that show exemplars of both the linear combination fitting and Gaussian peak fitting that were used to determine the sulfate composition and relative abundance of each oxidation state, respectively.

Figures S1 and S2 were added to pages 4 & 5 of the Supplementary Information.

Page 6, line 34: The primary emission samples were only characterized at the bulk level. Given that most of the S(0) in the ambient samples was observed only for individual particles, wouldn't this lead to wrong conclusions regarding their comparability?

Reviewer 2 also wanted clarification on the use of bulk emission source data. Emission sources were only characterized at the bulk level, and this poses a problem for data interpretation. We still use the emission source data for comparison with bulk ambient aerosol data and to a limited extent the individual particle data to glean any possible insights; however, we will further clarify that emission sources were collected in the bulk mode and any comparisons between the bulk emission source data and the ambient aerosol individual particle data are speculative.

Text reflecting that the emissions were collected in only bulk mode was added to the Discussion (page 9-10).

Page 8, line 22: This sounds as if this study comes to the conclusion that organosulfates were not present. However, as far as I can see no organosulfate standard was measured, nor are there any comparisons to literature data, so how do the authors come to this conclusion?

In this manuscript, we postulate that organosulfates are not a major component of sulfate aerosol. We were able to account for the sulfate using our present database, which suggests that organosulfates likely do not account for a significant portion of the sulfate aerosol. We do, however, state that contributions near or below 10% could be below the detection limit of this method. Since this is approximately the level that organosulfates are hypothesized to occur, it makes sense that we are able to account for all sulfate present in the sample with our database, even if some small portion of organosulfate is present.

This text was added to the Discussion (page 9).

Table S1: It is clear from this table that measurements of individual particles were only taken for few of the collected samples, in particular at Fire Station 8. Is there any specific reason for that?

Individual particle spectroscopy was not collected for every sample in our collection. Individual particles were examined on samples other than those noted in Table S1; however, the particles did not generate useable spectra. Table S1 represents the number of spectra that were collected that could be used for further analysis.

Text indicating that the number of usable spectra are reflected in Table S1 was added to the Methods Section under Synchrotron-based Spectromicroscopy (page 5).

More information on the generation of the sulfur maps is needed, either in the Data Analysis section or at least in the caption of figure S1. For example: are the maps normalized? What is the unit of the scale bars?

Reviewer 2 also wanted more detailed information to assist with the interpretation of the multi-energy maps presented in Figure S1. For the multi-energy maps, the units are raw counts from the detector. At this beamline, units such as mass per area are not provided. A more intense signal, that is the more counts present, indicates greater concentrations of sulfur in that region of the sample. The maps were generated at the whiteline energies of S0 and S+VI. The energies referenced in the methods are the actual settings used during experiment; in other words, these energies are not calibrated with the monitor stick. To identify the correct whiteline energies for the multi-energy mapping, a S-NEXFS spectrum was collected for the particle of interest immediately before mapping. The corresponding whiteline energies for S0 and S+VI were then taken directly off this spectrum. Individual maps were then collected at the whiteline energies determined for S0 and S+VI. Although it is possible that the energy drifted during mapping, the interval between the two measurements is short enough that this is not a problem. Due to the 6 ev difference in the whiteline energies of the S0 and S+VI oxidation states, we do not expect significant overlap between the two oxidation states in multi-energy mapping. Energy drift during mapping may reduce signal intensities of the S0 and S+VI oxidation states but the overall distribution patterns of the sulfur oxidation states in an individual particle should remain relatively unaffected.

Fitting was not done to determine the relative abundance of sulfur at each oxidation state for each pixel. Instead, more accurate data from individual particle spectroscopy was collected and presented in the paper and supplement.

This text was added to the Methods Section under Synchrotron-based Spectromicroscopy (page 6).

Technical corrections:

page 5, line 35: "monochrometer" should be "monochromator"

This was addressed (page 6).

page 6, line 8: "raster" should be "rastered"

This was addressed (page 6).

---

## Author Response (AR2)

**Author Response:**

**We thank these reviewers for their very helpful comments and suggestions.**

**Comments from Referee #1**

This paper discusses the composition and the oxidation state of sulfur in atmospheric aerosols collected in the general Atlanta area. The overall work is of interest to the community and is well written and has adequate discussion regarding the basic results. Some better presentation of the data and results and clarification of data processing methods however, should be included in at least the supplemental materials.

**Specific Comments:**

page 5, line 15: I very much like the addition of the sulfate standard database in the supplemental material. However, can the authors address the potential of self-absorption effects of sulfate standards? Particularly as this may possibly contribute to broadness and amplitude reduction of some of the standard peaks? Can the authors elaborate more on whether these data were collected in the bulk or microscale mode? Any more info on the standards particle size, more than "homogenized"?

Sulfur standards were ground using an agate mortar and pestle to the consistency of a fine talcum powder (approximately 10 microns). A cellulose acetate filter was gently dredged through a small quantity (less than 1 mg) of powder placed on a microscope slide. This procedure produced a thin and almost imperceptible coating on the filter in order to limit the thickness and thus self-absorption. S-NEXFS spectra of sulfate standards were collected in bulk mode. Self-absorption must be carefully controlled when measuring fluorescent X-rays from thick specimens; however, the effects of self-absorption are limited to the region of the spectrum above the K-edge (Iida and Noma, 1993; Bajt et al., 1993). In our repeated measurements, the post-edge features were consistent and reproducible which allows us to distinguish between sulfate standards.

This text was added to the Methods section under Sulfate Standards (page 5).

Supplemental material should at least include some plots of NEXFS spectra and their fit results of some typical sample to be able to evaluate the data quality and method of fitting. Both types of fitting with the linear combination of sulfate standards and the use of Gaussian functions should be shown.

Additional figures were added to the supplement that show exemplars of both the linear combination fitting and Gaussian peak fitting that were used to determine the sulfate composition and relative abundance of each oxidation state, respectively.

**Figures S1 and S2 were added to pages 4 & 5 of the Supplementary Information.**

page 5, line 34: The authors follow a rigorous method of following the monochromator energy drift. However, they do not reference what was mono energy was calibrated with initially? This information is critical to allow comparison to any other published dataset of sulfur spectroscopy.

The energy was calibrated using an elemental sulfur standard ( $S^0$ ) measured at beamline 2-ID-B. The whiteline energy of the elemental sulfur standard was aligned to 2472 eV (Cozzi et al., 2009). All subsequent data uses 2472 eV as the reference energy for  $S^0$  during the data alignment mentioned by Reviewer 1. Furthermore, for every measurement two spectra are collected: the specimen and an aluminum sulfate standard on the monitor stick. This approach means that all spectra are referenced to the aluminum sulfate standard and the initial calibration is not as crucial on this beamline.

**This text was added to the Methods section under Synchrotron-based Spectromicroscopy (page 5-6).**

Was there any calibration of the sulfur concentration in the samples? If so, was this a theoretical calculation or an empirical calibration. It would be useful to discuss not only the relative changes in oxidation states, but the overall concentrations observed as well. This information would help in examining the various sources.

**The concentrations of sulfur were not measured in this study (see response for next comment below).**

Fig S1. I would like to see more information regarding the multiple energy maps. What are the units of the map for each sulfur oxidation state? How were the units determined? Is the intensity here the intensity of the sulfur at each of the white line energies, or was a fit done to calculate the various proportions of each oxidation state? The latter would be a much more rigorous method, as there could be intensity of S0 and SVI is not completely unique to each of the white lines – that is there are contributions of each species to each of the measured energies. A proper method would be to measure the intensities at least N+1 energies (given S0 and SVI, N=2) and do a linear combination fit to determine the N species present at each map pixel. The choice of white line energies compared to the standard library seems off – there white line in the library is at  $\sim$ 2483 eV for sulfate typically, whereas the maps were stated to have been measured at 2480 eV. One would expect, based on the apparent energy calibration, that S0 would appear at 2474 eV. Given the 2480 energy, one may observe some significant intensity of sulfates such as the ammonium sulfates around the energies of S0. Could this mislead the interpretation of S0 in the multiple energy maps if a careful fitting was not performed?

For the multi-energy maps, the units are raw counts from the detector. At this beamline, units such as mass per area are not provided. A more intense signal, that is the more counts present, indicates greater concentrations of sulfur in that region of the sample. The maps were generated at the whiteline energies of  $S^0$  and  $S^{+VI}$ . The energies referenced in the methods are the actual settings used during experiment; in other words, these energies are not calibrated with the monitor stick. To identify the correct whiteline energies for the multi-energy mapping, a S-NEXFS spectrum was collected for the particle of interest immediately before mapping. The corresponding whiteline energies for  $S^0$  and  $S^{+VI}$  were then taken directly off this spectrum. Individual maps were then collected at the whiteline energies determined for  $S^0$  and  $S^{+VI}$ . Although it is possible that the energy drifted during mapping, the interval between the two measurements is short enough that this is not a problem. Due to the 6 eV difference in the

whiteline energies of the  $S^0$  and  $S^{+VI}$  oxidation states, we do not expect significant overlap between the two oxidation states in multi-energy mapping. Energy drift during mapping may reduce signal intensities of the  $S^0$  and  $S^{+VI}$  oxidation states but the overall distribution patterns of the sulfur oxidation states in an individual particle should remain relatively unaffected.

Fitting was not done to determine the relative abundance of sulfur at each oxidation state for each pixel. Instead, more accurate data from individual particle spectroscopy was collected and presented in the paper and supplement.

This text was added to the Methods Section under Synchrotron-based Spectromicroscopy (page 6).

**Technical Corrections:**

page 5, line33: Did the experiment actually use Vortex SDD with 5 mm2 area? This seems very small for standard XRF analysis. The more standard value for analysis is 50 mm2.

The reviewer is correct that the area of the Vortex SSD was  $50 \text{ mm}^2$  (page 6).

page 9, line 5: Text refers to Figure S4. This does not exist. Do the authors mean Figure S1?

Text referring to Figure S4 was corrected (page 9).

**Comments from Referee #2**

**General comments:**

This paper presents the results of a study on the speciation of sulfur in ambient aerosol samples from the greater Atlanta area. The topic is of interest to this journal and studied with an appropriate technique. The article is generally well written, but the description of data collection and processing needs to be improved before publication.

**Specific comments:**

Page 3, line 27: Given that secondary sources are discussed for reduced sulfur and the mention of organosulfates in the Discussion section, I think a short sentence on secondary S(+VI) would be warranted.

A short sentence was added to the Discussion about the possibility of secondary formation of sulfates. Using just these techniques, it is unclear what the role of secondary sulfates are in these ambient samples; however, the high concentrations of metal sulfates suggests that secondary processes do influence the composition of S(+VI) in ambient aerosols.

**This text was added to the Discussion (page 9).**

Page 5, line 13: I commend the inclusion of the standards database in the supplementary information. I do however think that more information on the nature of the measured standards (e.g. particle size) would vastly improve the usefulness of this database.

Sulfur standards were ground using an agate mortar and pestle to the consistency of a fine talcum powder (approximately 10 microns). A cellulose acetate filter was gently dredged through a small quantity (less than 1 mg) of powder placed on a microscope slide. This procedure produced a thin and almost imperceptible coating on the filter in order to limit the thickness and thus self-absorption.

This text has been added to the manuscript in the Methods section under Sulfate Standards (page 5).

Page 5, line 29: the description of the settings doesn't add up properly to me. It states that a 50 eV range was scanned in 0.33 eV steps with a dwell time of 1 s per step. If three full spectra were collected per particle or area (which is how I'd interpret the following sentence), how can the total dwell time be only 3 s? Also, given that multiple spectra were collected for each particle, did you observe any indication of beam damage (especially for S(0))?

Dwell time is used to refer to the time spent at each step in the S-NEXFS spectrum. In this case, because three spectra with a 1 s dwell time each were often necessary to get enough signal for a high quality spectrum, the effective dwell time was at least 3 s because a minimum of 3 s were spent at each energy step in the spectrum. With the 50 eV range and 0.33 eV stepsize used in this study, measurements were taken at 150 energies per scan. For 150 energy steps with an effective dwell time of 3 s, the total time required to generate a S-NEXFS spectrum was approximately 7.5 minutes. We did oxidation tests on samples where we measured the same

particle repeatedly, creating effective dwell times of more than 10 s. Even with this longer dwell time per energy step, we saw no noticeable shift in oxidation state or the relative abundance of the oxidation states.

Dwell time was defined in the manuscript in the Methods under Synchrotron-based Spectromicroscopy (page 5-6).

Page 5, line 34: It is stated how a potential drift in energy calibration could be monitored. However, no mention is made of the original method of energy calibration. Was a sulfur standard used and if so, which one?

The energy was calibrated using an elemental sulfur standard (S0) measured at beamline 2-ID-B. The whiteline energy of the elemental sulfur standard was aligned to 2472 eV (Cozzi et al., 2009). All subsequent data uses 2472 eV as the reference energy for S0 during the data alignment mentioned by both reviewers. Furthermore, for every measurement two spectra are collected: the specimen and an aluminum sulfate standard on the monitor stick. This approach means that all spectra are referenced to the aluminum sulfate standard and the initial calibration is not as crucial on this beamline.

This text was added to the Methods section under Synchrotron-based Spectromicroscopy (page 5-6).

Page 6, line 31: Some spectra and the associated fits should be shown so that readers are able to evaluate data and fit quality (if nor here, then at least in the Supplementary Information).

Figures were added to the supplement that show exemplars of both the linear combination fitting and Gaussian peak fitting that were used to determine the sulfate composition and relative abundance of each oxidation state, respectively.

Figures S1 and S2 were added to pages 4 & 5 of the Supplementary Information.

Page 6, line 34: The primary emission samples were only characterized at the bulk level. Given that most of the S(0) in the ambient samples was observed only for individual particles, wouldn't this lead to wrong conclusions regarding their comparability?

Reviewer 2 also wanted clarification on the use of bulk emission source data. Emission sources were only characterized at the bulk level, and this poses a problem for data interpretation. We still use the emission source data for comparison with bulk ambient aerosol data and to a limited extent the individual particle data to glean any possible insights; however, we will further clarify that emission sources were collected in the bulk mode and any comparisons between the bulk emission source data and the ambient aerosol individual particle data are speculative.

Text reflecting that the emissions were collected in only bulk mode was added to the Discussion (page 9-10).

Page 8, line 22: This sounds as if this study comes to the conclusion that organosulfates were not present. However, as far as I can see no organosulfate standard was measured, nor are there any comparisons to literature data, so how do the authors come to this conclusion?

In this manuscript, we postulate that organosulfates are not a major component of sulfate aerosol. We were able to account for the sulfate using our present database, which suggests that organosulfates likely do not account for a significant portion of the sulfate aerosol. We do, however, state that contributions near or below 10% could be below the detection limit of this method. Since this is approximately the level that organosulfates are hypothesized to occur, it makes sense that we are able to account for all sulfate present in the sample with our database, even if some small portion of organosulfate is present.

**This text was added to the Discussion (page 9).**

Table S1: It is clear from this table that measurements of individual particles were only taken for few of the collected samples, in particular at Fire Station 8. Is there any specific reason for that?

Individual particle spectroscopy was not collected for every sample in our collection. Individual particles were examined on samples other than those noted in Table S1; however, the particles did not generate useable spectra. Table S1 represents the number of spectra that were collected that could be used for further analysis.

Text indicating that the number of usable spectra are reflected in Table S1 was added to the Methods Section under Synchrotron-based Spectromicroscopy (page 5).

More information on the generation of the sulfur maps is needed, either in the Data Analysis section or at least in the caption of figure S1. For example: are the maps normalized? What is the unit of the scale bars?

Reviewer 2 also wanted more detailed information to assist with the interpretation of the multi-energy maps presented in Figure S1. For the multi-energy maps, the units are raw counts from the detector. At this beamline, units such as mass per area are not provided. A more intense signal, that is the more counts present, indicates greater concentrations of sulfur in that region of the sample. The maps were generated at the whiteline energies of S0 and S+VI. The energies referenced in the methods are the actual settings used during experiment; in other words, these energies are not calibrated with the monitor stick. To identify the correct whiteline energies for the multi-energy mapping, a S-NEXFS spectrum was collected for the particle of interest immediately before mapping. The corresponding whiteline energies for S0 and S+VI were then taken directly off this spectrum. Individual maps were then collected at the whiteline energies determined for S0 and S+VI. Although it is possible that the energy drifted during mapping, the interval between the two measurements is short enough that this is not a problem. Due to the 6 ev difference in the whiteline energies of the S0 and S+VI oxidation states, we do not expect significant overlap between the two oxidation states in multi-energy mapping. Energy drift during mapping may reduce signal intensities of the S0 and S+VI oxidation states but the overall distribution patterns of the sulfur oxidation states in an individual particle should remain relatively unaffected.

Fitting was not done to determine the relative abundance of sulfur at each oxidation state for each pixel. Instead, more accurate data from individual particle spectroscopy was collected and presented in the paper and supplement. This text was added to the Methods Section under Synchrotron-based Spectromicroscopy (page 6).

Technical corrections:

page 5, line 35: "monochrometer" should be "monochromator"

This was addressed (page 6).

page 6, line 8: "raster" should be "rastered"

This was addressed (page 6).

**Composition and oxidation state of sulfur in atmospheric particulate matter**

Amelia F. Longo1, David J. Vine2, Laura E. King1, Michelle Oakes3, Rodney J. Weber1, L. G. Huey1, Armistead G. Russell1, Ellery D. Ingall1

5 1School of Earth and Atmospheric Sciences, Georgia Institute of Technology, 311 Ferst Drive, Atlanta, GA 30332-0340, USA

2Advanced Photon Source, Argonne National Laboratory, 9700 S. Cass Avenue, Argonne, IL 60439, USA
 3Tennessee Department of Environment and Conservation, Division of Air Pollution Control, Nashville, TN 37206, USA

1

10 Correspondence to: Ellery Ingall (ingall@eas.gatech.edu)

**Abstract.** The chemical and physical speciation of atmospheric sulfur was investigated in ambient aerosol samples using a combination of Sulfur Near-Edge X-ray Fluorescence Spectroscopy (S-NEXFS) and X-ray fluorescence (XRF) microscopy. These techniques were used to determine the composition and oxidation state of sulfur in common primary emission sources and ambient particulate matter collected from the greater Atlanta area. Ambient

- 5 particulate matter samples contained two oxidation states:  $S_{\bullet}^{0}$  and  $S_{\bullet}^{+VI}$ . Ninety-five percent of the individual aerosol particles (> 1 µm) analyzed contain  $S_{\bullet}^{0}$ . Linear combination fitting revealed that  $S_{\bullet}^{+VI}$  in ambient aerosol was dominated by ammonium sulfate as well as metal sulfates. The finding of metal sulfates provides further evidence for acidic reactions that solubilize metals, such as iron, during atmospheric transport. Emission sources, including biomass burning, coal fly ash, gasoline, diesel, volcanic ash, and aerosolized Atlanta soil, and the commercially
- 10 available bacterium *Bacillus subtilis*, contained only  $S_{\star}^{+VI}$ . A commercially available *Azotobacter vinelandii* sample contained approximately equal proportions of  $S_{\star}^{0}$  and  $S_{\star}^{+VI}$ .  $S_{\star}^{0}$  in individual aerosol particles most likely originates from primary emission sources, such as aerosolized bacteria or incomplete combustion.

[revised manuscript text omitted]

|                | Deleted: zinc          |
|----------------|------------------------|
| -1             | Deleted: e             |
| $\overline{)}$ | Deleted: e             |
| r
T         | Deleted: .             |
|                | Deleted:               |
|                | Formatted: Superscript |

[revised manuscript text omitted]